# Reducing the Physical Burden of Older Persons’ Household Caregivers: The Effect of Household Handrail Provision

**DOI:** 10.3390/ijerph19042272

**Published:** 2022-02-17

**Authors:** Ruttana Phetsitong, Patama Vapattanawong

**Affiliations:** 1Faculty of Physical Therapy, Mahidol University, Nakhon Pathom 73170, Thailand; ruttana.phe@mahidol.ac.th; 2Institute for Population and Social Research, Mahidol University, Nakhon Pathom 73170, Thailand

**Keywords:** physical burden, caregiver, older person, household, handrail, Thailand

## Abstract

The household handrail is necessary for dependent older people to perform their daily living activities, improve caregiving competency, and reduce caregiver burden. This study aimed to explore physical burden levels and examine their association with handrail provision among caregivers in older people’s households in Phuttamonthon District, Thailand, in 2017. This study used the physical dimension of the Caregiver Burden Inventory to quantify the levels of physical burden among 254 caregivers in households with a dependent older person. It classified the studied households into three categories: no handrail, one handrail, and more than one handrail. The analysis employed the ordinal logistic model approach. The findings showed that the mean physical burden score was 5 ± 3.85, indicating a high burden. After adjusting for potential factors, the caregivers in older people’s households with one handrail were less likely to experience a high physical burden than those without a handrail (OR = 0.30, 95% CI = 0.14–0.67). Nonetheless, the analysis found no significant differences in physical burden between caregivers of households with more than one handrail and those of households with no handrails. Having handrails in housing might enhance older people’s ability to adjust to disability and illness, ultimately reducing the physical care burden of caregivers. However, having the appropriate number of handrails in older people’s households should be considered.

## 1. Introduction

The world’s population is getting older, accompanied by an increase in age-related disability [1]. For disabled older people, user-friendly housing is a basic need for them and their caregivers. According to the International Classification of Functioning, Disability, and Health, known as the ICF framework, facilitating environments, including home modifications and assistive devices (i.e., handrails and grab bars), helps to maintain older people’s independence and makes caregiving for older people with impairments easier and safer [2,3]. Therefore, the provision of household handrails may be a possible determinant that influences the quality of life of older people and appears to reduce the burden of caregivers in providing support for the activities of daily living (ADLs) of older people [4,5].

A caregiver is a key to the successful care of older people. Caregiver characteristics and caregiving environments can contribute to older persons’ health outcomes. Care provision at home by persons who lack care training or resources and supportive environments can generate high physical health problems, significant fatigue, and caregivers’ sleep impairment [6,7,8,9]. The potential of the physical burden as a result of caregiving is a factor that can affect the termination of a caregiver’s role or, in the worst cases, elder abuse [10].

In Thailand, like other Asian societies, the provision of healthcare services for older citizens is still dependent on home-based care [11,12,13,14]. With their sense of connectedness with home and concern for security, most of the Thai elderly prefer to live at home. The important issue of concern is that the caregiver burden tends to be an increasing problem and presents a growing challenge for Thailand’s aging society as a falling potential support ratio has been identified [15]. Furthermore, Thai caregivers are growing older. These trends will increase the physical burden among caregivers due to their age, which is related to physical stress and morbidity [1]. Accordingly, the bulk of the burden of long-term care of older Thai persons falls upon the caregiver. Thus, it is important to focus on the physical caregiver burden.

The concept of home modification has evolved over time. It was initially defined as “changes to the structure of the dwelling (e.g., widening doors, adding ramps, providing better accessibility) and the installation of assistive devices inside or outside the dwelling (e.g., handrails, grab bar, lifts, etc.), to help people to be more independent and safer in their own home and reduce any risk of injury to their carers and care workers” [16].

Effects of home modifications and caregiving appear to be broad and still inconsistent [17,18]. Positive effects have been reported. Home modifications reduce the amount of care provided [18]. Household handrails are considered a necessity for dependent older persons in performing their ADLs and can also improve the competency of caregiving, as well as reduce caregiver burden [19,20,21]. However, existing theoretical models described assistive devices are unable to replace caregivers for most older people living with disabilities in the community. Instead, assistive devices appear to supplement personal care and seem to be associated with a higher probability of taking more care hours and more caregivers in assistance with more ADL tasks [22]. Although adaptation in the home can better support older people to remain independent for longer, the number of modifications has received attention [23].

Based primarily on the stress theory [24,25,26], there are a number of factors that could affect physical caregiver burden. The physical condition of older caregivers is likely to be inferior to that of younger caregivers (e.g., because of chronic diseases) [26]. The heavy care tasks and the necessity of frequently assisting the person they care for in their activities of daily living, alongside their accumulated stress, might have various negative effects on caregivers [24]. Based on older person characteristics, handrails may be beneficial for an older person with an expanded fall chance due to physical disability, however they may be possibly ineffectual for someone with cognitive impairment [27].

Long-term care and caregiver burden studies have devoted attention to individual characteristics of caregivers and older people in their analysis and lack evidence of how household factors such as handrail provision affects physical caregiver burden. Therefore, the purpose of this study was to explore the level of the physical burden of caregivers in older persons’ households and to examine the association between household handrail provision and the physical burden of the caregiver in older persons’ households. Specifically, this study aimed to test the following hypotheses: First, mean physical caregiver burden will be as high as that presented of the falling potential support ratio of Thailand, and second, there will be significant differences in physical burden between caregivers of households without a handrail and households with one handrail or those with more than one handrail. Given the rapidly aging population, understanding the household factors that play an important role in mitigating physical caregiver burden could inform the development and monitoring of long-term care policies.

## 2. Materials and Methods

### 2.1. Study Design, Population, and Data Collection

A cross-sectional study was employed. The data were drawn from our earlier research project conducted in 2017 in Phutthamonthon District, Nakhon Pathom Province, Thailand, where the western suburbs of Bangkok are located (Figure 1). It was carried out in 254 private households of older people who were aged 60 years and over and had a caregiver providing care in the home.

Phutthamonthon has a strong Buddhist-based ideological system. It has long been known for its kin-based agriculture and animal farming economy, but it has now been industrialized and is a more urban environment, resulting in a change in demographic trends toward smaller family size and more female employment outside the home [28]. The Phutthamonthon recorded the largest percentage increase in the number of households for the province [29]. In addition, chronic disease risk factors and health problems of populations in such settings of social upheaval are high [30].

The inclusion criteria for being a primary caregiver eligible for the study were as follows: Having a primary role in providing day-to-day care with ADLs for the target older person at home. In Thailand, home-based care is delivered by both informal carers and formal caregivers. However, the informal caregiver is the first preference for older people. From a practical standpoint, different care forms can mix together. It is critical to understand how household handrails can minimize the physical care burden of older people’s caregivers, regardless of whether a caregiver is informal or not. Therefore, this included both informal caregivers (any family member, partner, relatives, friends, or neighbors who had a significant personal relationship with the older household members), and formal caregivers (a provider associated with a formal service, whether a paid worker or a volunteer); being at least 15 years of age (to gain information based on the caregiving context and caregiver’s perspective); and willing to participate in the study.

Before the actual data collection, a list of households with older persons was compiled. A total of 4056 households with older people in three sub-districts (Salaya, Mahasawasdi, and Klong Yong) were identified by the medical records officers of the Phutthamonthon District Hospital and the Sub-district Health Promoting Hospitals. Among the 4056 households, 363 were identified as having an older person(s) who needed care, and this information was verified by home visiting teams and the village volunteers from each local hospital. However, after screening with a household questionnaire, it was evident that a total of 319 households actually had an older person(s) who needed a caregiver. Of the 319 households approached, 30 households were allocated for pre-test measurement of questionnaire reliability, 19 households were excluded based on the lack of consent to participate, and 16 households were excluded due to an incomplete questionnaire. Ultimately, 254 households were included for the data analysis (Figure 2).

Data were collected from the eligible private households of 18 villages of three sub-districts (Salaya, Mahasawasdi, and Klong Yong) of Phuthamonton District in August 2017. The interviewer teams were trained for data collection. Two interviewers were assigned to each eligible household. One interviewer interviewed the responsible adult using the household questionnaire, and the primary caregiver of the household using the caregiver questionnaire. Another interviewer interviewed the target older person with the older-person questionnaire. The data collection took about 40 min. In cases where an interview was not completed in one visit, follow-up visits were arranged up to a maximum of three.

The older household members were asked to identify their primary caregivers in order to recruit them into the study. As a primary caregiver of the target older person, the caregiver’s status was ascertained by the question “Are you the person who most regularly provides care to the target older person?” In cases where more than one caregiver met the criteria, the primary caregiver was selected hierarchically based on the following criteria (in order): (i) they were involved in the older person’s treatment, and (ii) they were contacted in case of emergency.

### 2.2. Physical Caregiver Burden

Caregiving for an older person with ADL difficulties presents burdens across physical, psychological, social, and financial dimensions [7]. For this study, physical caregiver burden was defined as a primary caregiver having physical reactions to the imbalance of demands and resources placed on them by various factors, including the socio-demographic characteristics of both the caregiver and the older person and the health conditions of the older person as well as their environment. This was assessed by the physical dimension of the Caregiver Burden Inventory (CBI). The original CBI has been validated in many languages (e.g., Chinese, Italian, Brazilian, Spanish) and it has favorable reliability (0.77–0.94) [31,32,33,34,35]. There are four items for the physical dimension as follows: “I am not getting enough sleep”, “My health has suffered”, “Caregiving has made me physically sick”, and “I am physically tired”. For each item, the scale’s response options consisted of 0 (never), 1 (rarely), 2 (sometimes), 3 (quite frequently), and 4 (nearly always), where the respondents were asked to indicate which score best represented how often the statement described the caregiver’s feelings while taking care of the older person. Thus, the score for the domain of physical burden ranged from 0 to 16. The cut-off scores of the physical burden subscale in this study were determined relative to the original cut-off value. Summed scores higher than a quarter of the total score indicated a high caregiver burden [31]. Therefore, the interpretation of the total score was 0 = “no physical burden”, 1–4 = “lower physical burden”, and 5–16 = “higher physical burden”. The CBI was originally written in English. It was translated into Thai for this study. The initial translation (or forward translation from the original English to Thai) was done by two independent translators. Then, a translation back from the Thai language into English was made to verify the accuracy of the Thai version. In the last step, native users of English compared the original English version and the back-translated English versions to confirm equivalence or detect discrepancies. These were resolved to produce a pre-final version of the translated CBI, and the internal consistency of the physical dimension in this study was found to be reasonable (Cronbach’s alpha = 0.80).

### 2.3. Household Handrail Provision

Research literature has used the term ‘handrail’ to define a narrow rail for grasping with the hand as support both inside and outside the home. They can be installed on the stairs, in the bathroom or toilet, and so on [36,37,38,39]. This study defined household handrail provision as having a stair handrail, a handrail in the bedroom, or a handrail in the bathroom/toilet of the older person’s household. The sample households were classified into three types: 1 = households with no handrails, 2 = households with a handrail in one place, and 3 = households with handrails in more than one place.

### 2.4. Covariates

Potentially important factors were included in an analysis to control variables that could affect the primary outcome of this study and to increase the efficiency of the analysis, thus producing more precise and stronger statistical values for an effect. There are two main classical theoretical frameworks that are widely used in the empirical literature of caregiver burden to guide the identification of related factors affecting burden [40,41,42,43,44]. Most of the burden studies related to caregivers were primarily driven from the original conceptualization of the Pearlin stress process [24] that was developed from the transactional stress theory and coping by Lazarus and Folkman in 1984 [45], while the stress appraisal model of Yates [25] focused on the appraisal of stressors and available resources for caregivers. Following these two stress models [24,25], four domains play important roles in explaining variables of a physical caregiver burden: (i) background characteristics, (ii) older people’s health status, (iii) caregiving hours, and (iv) social support.

According to household background variables, living arrangements and economic household factors have been cited as a predictor of a stressor or strain [24]. Living arrangement refers to the co-residence of the target older person with their caregiver and other household members. Four different living arrangements of the older person(s) were defined: (i) living alone or living together, (ii) living with only a caregiver, (iii) living with a caregiver and others, and (iv) living with others but not with a caregiver. The living arrangement was constructed based on the information about household members living in the household combined with a question asked of the primary caregiver: “Do you live in the same household with the older person?” Relative household wealth is a proxy of the socio-economic status of the household. The wealth index was measured based on a set of household characteristics and ownership of assets. These attributes were converted into a wealth index by the principal component analysis [46] and converted into relative household wealth quintiles (1 = poorest, 2 = poor, 3 = medium, 4 = rich, and 5 = richest). The index combined data on 14 indicator variables including possessing a CD/DVD/VCD player, a microwave oven, a washing machine, an air conditioner, an electric water heater in the bathroom, a mobile phone, a computer, a tablet computer, a car/small truck/pick-up truck/van/farm tractor, a motorcycle, a two-wheel tractor, own household land, access to safe drinking water (treated tap water, bottled water, or water from a vending machine), and access to a safe water supply (tap water, treated tap water, bottled water, or water from a vending machine). It is important to note that wealth quintiles from the primary data collection only represent the distribution of wealth among the 254 eligible households of Phutthamonthon District, Nakhon Pathom Province.

Caregiver background variables were used to capture possible determinants of physical caregiver burden, including age, sex, marital status, educational attainment, working status, the relationship of the caregiver to the older person, and duration of care for the older person. When a caregiver is a primary carer for dependent children in addition to an older or disabled household member, they are referred to as a “sandwich carer”.

Regarding the stressors relating to caregiving, older people’s health status, including functional dependency, behavioral problems, and cognitive impairment, have been repeatedly mentioned in the literature. For this study, functional dependency was defined as the inability to perform ADLs independently. The level of functional dependency was based on assessing ten ADLs using the Barthel’s Index Scale (Thai version). These ADLs included feeding, grooming, transferring, toilet use, mobility, dressing, stairs, bathing, bowels, and bladder. The score was determined by guidelines of the Thai Institute of Geriatric Medicine (≥12 = independent; 5–11 = partial dependency; and ≤4 = dependent, out of 20 possible points). Cognitive impairment was that of decreased intellectual ability leading to the incapacity of an individual to manage their social or occupational activities. For this study, the Mini-Mental State Examination (Thai version; MMSE-Thai 2002) was used. If the summed score was less than the cut-off score, the older person was determined to have cognitive impairment. The score categories were based on guidelines from the Thai Institute of Geriatric Medicine (14 out of 23 points for illiterate; 17 out of 30 points for primary school level; and 22 out of 30 points for higher than primary school level). Behavioral problems were the presence of behavioral challenge(s) based on the NeuroPsychiatric Inventory-Questionnaire (NPI-Q Thai) self-administered questionnaire, which was completed by caregivers about the older person in their care. There are 12 questions that include the presence of delusions, hallucinations, agitation/aggression, depression, anxiety, irrational euphoria, apathy, disinhibition, irritability, motor disturbance, disruptive nighttime behavior, and adverse eating and appetite changes. The response to each question is “yes” (present) or “no” (absent or not applicable). Having at least one of these symptoms indicates the presence of the behavioral problem(s).

The appraisal of caregiving is affected by the older person’s health status, and it can be measured by the number of hours of caregiving [25]. In this study, caregiving hours were the time spent on assisting the older person with routine daily activities including feeding, grooming, bathing, dressing, toileting, mobility, and/or transfers by the primary caregiver. This study categorized the number of caregiving hours a day into three levels: low = 0–1 h, medium = 1.1–3 h, and high ≥3 h [47].

Social support is a significant and direct mediator of caregiver burden and well-being [25,48]. It refers to help or support for the primary caregiver when they are having difficulty in providing care for the older person. This variable was measured as the perception of the primary caregiver of available assistance. Support may come from household members, family outside the household, friends, neighbors, community leaders, groups, organizations, and foundations. The measurement scale included “0 = not received” versus “1 = received”.

### 2.5. Statistical Methods

All statistical analyses were carried out using STATA/SE 14.0 [49]. The value for alpha was set at 0.05 for statistical significance; all tests were two-tailed. Descriptive analysis was performed to summarize the distribution of the background variables and to explore the quantitative level of physical caregiver burden. Ordinal logistic regression models were applied to predict the relationship between household handrail provision and physical caregiver burden. The models not only account for the ranked response inherent in ordinal scales, but can also adjust for confounding and determine effect modification from a modest sample size [50,51]. The model analysis of this study is based on combining theoretical frameworks on the caregiving stress process of Pearlin [24] and Yates [25], as shown in Figure 3.

Therefore, six model analyses were performed: Model 1 was an unadjusted model fitting dependent and independent variables only; Model 2 was a caregiver background adjusted model; Model 3 was a caregiver and household backgrounds adjusted model; Model 4 was a caregiver and household backgrounds plus older person’s health status adjusted model; Model 5 was a caregiver and household backgrounds plus older person’s health status and caregiving hours adjusted model. Model 6 was a fully adjusted model, which included all five important covariates: caregiver backgrounds, household backgrounds, older person’s health status, caregiving hours, and social support.

## 3. Results

A total of 254 older-person households were included in the study. The average household size of the households was 4.0 persons (SD = 1.6; range = 1–9). The majority of the older person(s) lived in the same household with their primary caregivers. The most common living arrangement was a household where the older person(s) lived with the caregiver and other household members (*n* = 192; 75.6%). The wealth index score was divided into five quintiles. Of 254 households, households in the first quintile (20.1%) were considered the poorest, whereas households in the fifth quintile (19.3%) were considered the richest. Again, it is important to note that the relative household wealth quintiles of this study only represent the distribution of wealth among the 254 households of the sample eligible for this study.

The 270 older individuals and 254 primary caregivers of the older-person households were identified. The mean age of the older people was 78.6 ± 9.1 with a range of 60–100 years old. Nearly half were in the oldest old group (47.8%). In total, 58.9% of the older people were female. The majority were widowed, with primary education. At the household level, more than half of older person’s households were households with at least one dependent older member. About half were households with an older person with dementia. In the lower half were households where at least one older person exhibited problematic behavior.

Regarding caregiver characteristics, 254 caregivers were non-older (aged <60 years old). The mean age of the caregivers was 54.6 ± 12.6 years with a range of 21 to 90 years. The vast majority (95.3%) were informal caregivers, and only 4.7% were formal caregivers. They were mostly females (*n* = 170; 83.5%), and most of the caregivers were kin, either the daughters or spouses of the older person they cared for. More than half (65.4%) of the caregivers were married or cohabiting. About half of the caregivers had only completed primary school. Around two-thirds of the caregivers had worked in another job while providing care to an older person(s). Nearly a quarter of the primary caregivers in this sample of households were sandwich carers. Most caregivers (57.8%) had provided caregiving for the older person(s) for more than four years, with an average duration of 6.5 years (±5.8).

Regarding caregiving hours or time spent assisting with ADLs, overall, the mean amount of time spent assisting with basic ADLs was 3.3 (±3.8) hours a day. The majority of caregivers spent more than four hours caregiving daily.

Nearly all (85.4%) received support from people and/or the home community (household members, family members, friends, neighbors, community leaders), whereas 14.6% of caregivers did not receive social support from either people or organizations when they had difficulty in caring for the older person. The caregivers in households with handrails were more likely not to receive social support from both people and organizations. However, the pattern of social support received was not significantly different between handrail and non-handrail households.

Overall, the findings showed that the majority of the households had no handrail (*n* = 110; 43.3%). There were 91 (35.8%) and 53 (20.9%) households with handrails in one place and with handrails in more than one place, respectively. Among 91 households with one handrail, there were 89 households where handrails were installed at the stair. Only one was in the bathroom or in the bathroom/toilet. The analysis indicated that the higher the socio-economic status, the more handrails there were in the households. Table 1 breaks down the socio-demographic backgrounds of the households, the living arrangements of older persons in the household, the caregivers’ socio-demographic characteristics, the health status of the older people, the caregiving hours, and the social support of the primary caregivers by household handrail provision.

### 3.1. Levels of Physical Caregiver Burden

Overall, the findings showed that the mean physical burden score was 5 (SD = 3.85) and ranged from 0 to 16. The interpretation of the Caregiver Burden Inventory score indicates a high level of physical burden among the caregivers of dependent older persons in Phutthamonthon. The proportion of the caregivers perceiving a high physical burden was about half (46.1%) of the total caregivers, whereas 37.0% and 16.9% of the caregivers perceived having a low level and no burden, respectively (Table 2).

### 3.2. Effects of Household Handrail Provision on Physical Caregiver Burden

In order to investigate the effect of household handrail provision on their perceived physical caregiver burden (1 = no burden, 2 = lower burden, 3 = higher burden) while controlling for other potentially important factors, it is essential to employ multivariate analysis based on ordinal logistic regression. However, after applying the Brant test, a significant test statistic provides evidence that the parallel regression assumption has been violated. Therefore, for this study, a generalized ordered logistic model with the autofit option proposed by Williams [52] was employed. The major advantage of generalized ordered logistic regression is that it is less restrictive than the parallel lines model estimated by an ordered logistic regression model. The autofit option greatly simplifies the process of identifying partial proportional odds models that fit the data.

The odds ratio for the ordinal logistic regression models evaluating the association between household handrail provision and physical caregiver burden is displayed in Table 3. Overall, the caregivers in older person’s households where there was a handrail in one place were less likely to experience a high physical care burden compared with the caregivers of the households without a handrail.

In Model 1 (unadjusted analysis), there was a significant difference in physical burden between caregivers of the one place-handrail households and the non-handrail households. The caregivers in the one place-handrail households had 0.56 times (95% CI = 0.33–0.94) less chance of physical burden than the non-handrail household caregivers.

After adjusting for caregiver background variables, including age, sex, marital status, education, working status, relationship to the older person(s), duration of being a caregiver, and sandwich carer status, Model 2 shows that the effect of household handrail provision on physical caregiver burden became non-significant (OR = 0.58, 95% CI = 0.32–1.04). In Model 3, which added the household background characteristics (household wealth index and living arrangement), it was found that the caregiver of the one place-handrail household remained statistically significant, but the size of the odds ratio was reduced. The caregivers of the one place-handrail household were less likely to suffer from a high physical burden than the caregivers of non-handrail households by 58% (OR = 0.42, 95% CI = 0.22–0.80).

For Model 4, which adjusted for the caregiver and household backgrounds and older person’s health status (i.e., cognitive impairment, functional dependency, behavioral problems), the findings show that the effect of household handrail provision on physical caregiver burden remained statistically significant. Compared to the adjusted Model 3, there was a lesser value of the odds ratio (OR = 0.36, 95% CI = 0.17–0.75). For Model 5, which added caregiving hours to the model, the odds ratios of the association were the smaller size (OR = 0.30, 95% CI = 0.14–0.66) compared to Model 4. These results were statistically significant.

In Model 6, which adjusted for all other potential factors, the significant level and odds ratios of the association were not changed from Model 5 (OR = 0.30, 95% CI = 0.14–0.67). However, the model suggests that the caregiver in a household that had handrails in more than one place was not likely to have a lower level of physical caregiver burden when compared to the caregiver of the household without handrails.

## 4. Discussion

According to the level of physical caregiver burden as measured by the CBI, about half of the caregivers of the older-person households in this sample in Phutthamonthon District of Nakhon Pathom Province in 2017 experienced a high burden. The level of the reported burden contrasts with previous studies of burden among well-educated caregivers of older people with dementia in clinical settings [53,54]. These are clinic-based studies in which most caregivers are well educated, have no financial problems, and receive strong social support [54,55,56,57]. However, outside the clinical setting, most Thai caregivers only have primary school education [58] and live in the home community. Moreover, direct comparison with those studies might be difficult. Cultural, religious, and spiritual beliefs play an important role in motivation for caregiving for dependent older members of the family [59,60,61]. The Thai “Bun-khun” (obligation) system is anchored in Buddhist caregiving principles. Buddhists believe that caregiving is repayment and expression of gratitude to their parents and other older persons who helped raise them [60,61]. Most residents of Phuthamonthon District are devout Buddhists, and the Buddhist belief and concept of seeking a path of moderation between two extremes might influence their engagement in caregiving [62,63]. Moreover, some coping strategies of Buddhism (e.g., prayer and meditation) might also mitigate the burden among caregivers in older-person households. Nevertheless, the prevalence of a high burden among caregivers of the households with dependent older persons in the study households is quite large. This suggests the need for some forms of respite care for these caregivers [54,64].

The results from multivariate analysis based on ordinal logistic regression indicated that caregivers in older person households with a handrail in one place were less likely to experience a high physical care burden compared with caregivers of a household without handrails. This might be because bathing, toileting, and stair handrails offer the potential to enhance the functionality of older people, resulting in an alleviation of the stress and sickness resulting from the physical care provided by caregivers [3,4,5].

Moreover, the findings highlighted the role played by household background characteristics (i.e., household wealth index and living arrangement) after adding these factors into the model. The adjusted association with the physical burden of caregivers was more significant, and the size of the odds ratio was reduced. Handrails in bedrooms, bathrooms, or on stairs are critical to most aging residents, but the older person households often lack these amenities, particularly the poor households, as presented in Table 1.

Based on the Pearlin stress process model [24], the moderator variable, such as social support, is a significant and direct predictor of caregiver burden. In contrast, the stress appraisal process model of Yates [25] did not find a direct effect of perceived social support on caregiver burden. The results of this study showed that the odds ratios of the association were still the same size as before adding the social support variable into the model. A potential explanation for this is that social support might be strongly associated with well-being rather than caregiver burden [65].

Although household handrails could enable older people to effectively engage within their environment and reduce the physical burden expressed by the caregiver, this study revealed that there was no significant difference in levels of physical burden between caregivers of households with handrails in more than one place and the non-handrail households. A likely reason for this is that handrails could not be substituted the assistance or support from a caregiver [22]. The more places of handrail installation at home, the greater the participation of the caregiver in older people’s specific self-care tasks [27]. Thus, the specific tasks participated by the caregivers for which the handrail might be used could cause a physical burden for them. The body mechanisms of using handrails in different places are different. To complete tasks with a handrail independently, older people need either sufficient muscles performance or full support or assistance from another. Naturally performing transferring movements while using a handrail requires body coordination, trunk, and lower limb supports [36]. For example, holding the toilet handrail and performing a sit to stand movement requires trunk and lower limb function to maintain the standing posture without releasing the hands from the handrail after standing up, or to pull the body in the forward direction and to raise it in the upward vertical direction. It is documented that if the older people incorrectly perform by grasping the handle using forearm pronation during using stairs or toilets, the handrail reaction force will be increased. This causes risks of older people falling and lower back disorder for the caregivers [39,66,67].

Furthermore, there is strong evidence that minor home modifications are a more effective strategy than many modifications for improving the performance of older people’s daily activities and caregiving by a caregiver [23]. In addition, caregivers also accept handrails in some places of the house [23,68]. They have different priorities and preferences in the use of handrails when they are in more than one place in the household. If the use of handrails requires more time or demands more energy from caregivers for many tasks, they might not be welcomed or accepted [3,68,69].

Other daily tasks such as household duties or commuting from home to work, on the other hand, may create other possible physical burdens, thus these variables must be monitored in the study. Additionally, because most older people live with their families, it is questionable whether other household members also serve as secondary caregivers.

With respect to the regression model analyses, the pseudo R^2^ statistics have been proposed as a measure of how well the ordinal responses can be predicted by a given logistic regression model and covariates. A pseudo R^2^ value is measured in the range of 0.0–1.0, approaching 0 as the quality of the fit of the model diminishes and 1 as the fit improves. However, the best pseudo R^2^ value is not universally reported [70]. The small sample size and numbers of variables could affect the sizes of the pseudo R^2^ [70,71]. In this study, although the low pseudo R2 was presented in the unadjusted model, the pseudo R^2^ of the adjusted models was good enough to explain the results, particularly in social science research [71].

The strength of this study is that it analyzed data at a household level. Nevertheless, there are a number of limitations, and recommendations for further study are described as follows: First, the study included only older persons in private households. Moreover, the data collection was carried out in one district of one province in central Thailand. Future research on older people and care burden should be broadened by including sample populations from other areas of the country. This would provide richer geographical and cultural comparisons. Second, regarding the analysis of factors affecting caregiver burden, this research was limited by its cross-sectional design. Therefore, a longitudinal design is recommended to investigate the patterns and true predictors of care burden for individuals over time. Third, in some context, other caregiver characteristics, such as income distribution and nationality, may also be likely to contribute to caregiver burden, further investigation is recommended. Finally, the caregiver burden is driven by mediators. The mediator such as coping strategy and coping resources (social support) can potentially block any point of the stress process, these two variables play a major role in explaining the variation of caregiver burden and well-being as they help caregivers to adapt better to their caregiving role [24,45]. Since home modification is one kind of coping strategy, this study only focused on the social support of the caregiver as a determinant of caregiver burden. A greater understanding of the coping mechanisms that a caregiver uses might help explain the findings from this research. Other strategies, such as seeking counseling and prayer, may decrease caregiver burden. This factor (i.e., coping) should be addressed in future research.

## 5. Conclusions

Physical caregiver burden tends to be an increasing problem and presents a growing challenge for aging societies as Thai caregivers are growing older. Building handrails in older people’s households is becoming more important. This study was conducted to address gaps in the previous studies on household handrail provision and physical caregiver burden in the context of Thailand. The prevalence of high physical caregiver burden among caregivers in older-person households with a need for care in Phutthamonthon District was quite large. This suggests the need for some form of respite care for these caregivers. The use of handrails in the household in one place by older people with disabilities appears to reduce the physical burden of their caregivers. However, this study does not provide support for the hypothesis that the use of a handrail in more than one place will reduce the burden. The results of this study provide a piece of crucial information for caregivers involved in long-term care. The results may also guide related social welfare institutions and government sectors in determining how to reduce physical caregiver burden by installing an appropriate number of handrails in the household of older people. 

## Figures and Tables

**Figure 1 ijerph-19-02272-f001:**
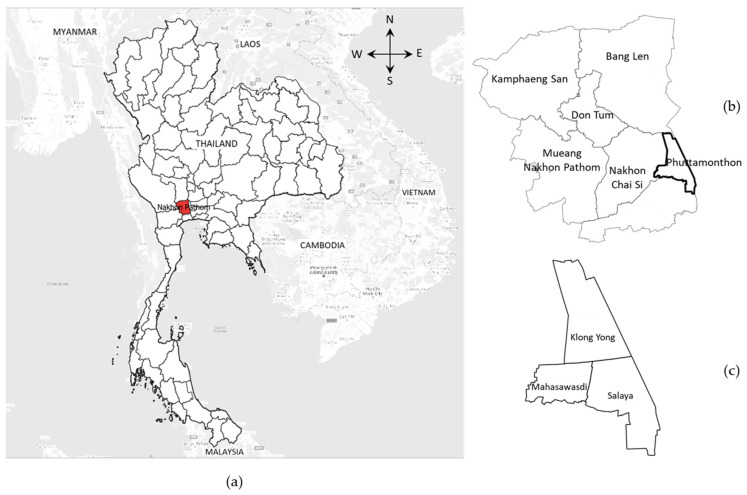
The study setting: (**a**) a map of Thailand (**b**) Phutthamonthon District, Nakhon Pathom Province of Thailand (**c**) three sub-districts of Phutthamonthon.

**Figure 2 ijerph-19-02272-f002:**
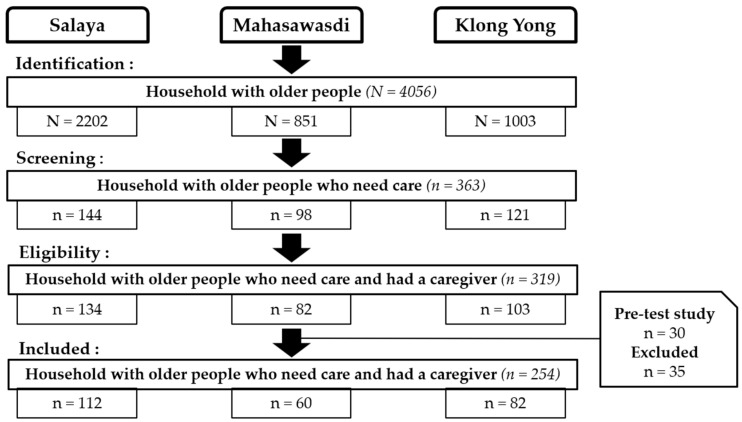
Flow chart of the population recruitment.

**Figure 3 ijerph-19-02272-f003:**
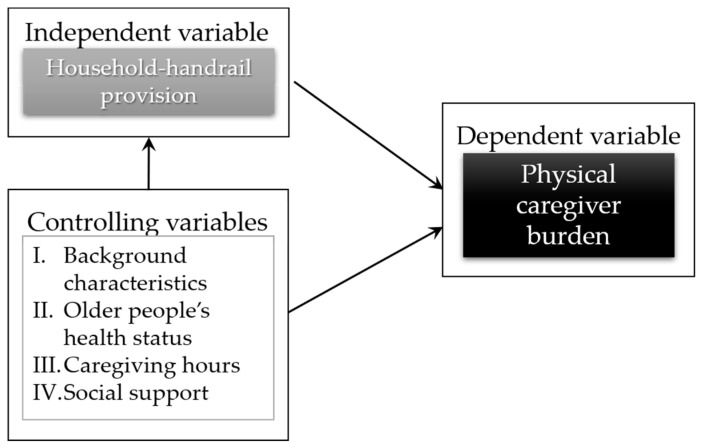
An analytical framework of this study.

**Table 1 ijerph-19-02272-t001:** Background characteristics by household handrail provision (*n* = 254 households).

		Household Handrail Provision		
Characteristics	All	No Handrails (%)	Having a Handrail in One Place (%)	Having Handrails in More Than One Place (%)
	*n* = 254	43.3	35.8	20.9
**Households**				
Living arrangement of older person				
lived alone/with another older person	12	66.6	16.7	16.7
lived with only their caregiver	29	48.3	34.5	17.2
lived with caregiver and others	192	41.1	36.5	22.4
lived with others but not a caregiver	21	42.8	42.9	14.3
Relative household wealth				
quintile 1, poorest	51	62.8	33.3	3.9
quintile 2, poor	50	58.0	26.0	16.0
quintile 3, medium	51	54.9	33.3	11.8
quintile 4, rich	53	28.3	45.3	26.4
quintile 5, richest	49	12.3	40.8	46.9
**Caregivers**				
Age mean (SD) (range: 21–90 years)	54.6 (12.6)	51.8 (13.3)	57.6 (10.0)	55.3 (14.1)
non-older	170	47.7	32.9	19.4
older	84	34.5	41.7	23.8
Sex				
male	42	52.4	30.9	16.7
female	212	41.5	36.8	21.7
Marital status				
married/cohabiting	166	43.4	37.3	19.3
single	42	33.3	42.9	23.8
widowed/divorced/separated	46	52.2	23.9	23.9
Educational level				
primary or less	153	42.5	42.5	15.0
secondary school	67	50.8	17.9	31.3
higher secondary	34	32.3	41.2	26.5
Working status				
working	174	44.8	36.8	18.4
not working	80	40.0	33.8	26.2
Relationship to older person				
kinship	240	43.3	36.7	20.0
non-kinship	14	42.9	21.4	35.7
Sandwich carer				
yes	63	49.2	28.6	22.2
no	191	41.4	38.2	20.4
Duration of care for older person (years)				
≤2	69	39.1	52.2	8.7
3–4	37	35.1	35.2	29.7
>4	148	47.3	28.4	24.3
**Health status of older persons**				
Functional dependency				
all independent	121	33.9	47.1	19.0
≥1 dependent	133	51.9	25.6	22.5
Cognitive impairment				
all absent	128	37.5	42.2	20.3
≥ 1 present	126	49.2	29.4	21.4
Behavioral problem				
all absent	155	45.8	34.2	20.0
≥1 present	99	39.4	38.4	22.2
**Caregiving hours**				
<1	78	38.4	43.6	18.0
1.1–3.0	97	49.5	28.9	21.6
>3	79	40.5	36.7	22.8
**Social support of caregivers**				
non-received	37	35.2	40.5	24.3
received	217	44.7	35.0	20.3

**Table 2 ijerph-19-02272-t002:** Levels of physical caregiver burden by household handrail provision (*n* = 254 households).

		Household Handrail Provision		
Physical Caregiver Burden	All	No Handrails	Having a Handrail in One Place	Having Handrails in More Than One Place
	*n* = 254	*n* = 110	*n* = 91	*n* = 53
Mean (SD)	5.0 (3.85)	5.3 (3.60)	4.3 (3.90)	5.7 (4.10)
Range	0–16	0–13	0–16	0–16
Level of the physical burden				
no	16.9%			
low	37.0%			
high	46.1%			

**Table 3 ijerph-19-02272-t003:** Odds ratio (OR) and 95% confidence interval (CI) from generalized ordinal logistic models evaluating the association between household handrail provision and physical caregiver burden (n = 254 households).

	Model 1		Model 2		Model 3		Model 4		Model 5		Model 6	
Physical Caregiver Burden	Unadjusted		Adjusted for Caregiver Background		Adjusted for Caregiver and Household Backgrounds		Adjusted for Caregiver and Household Backgrounds and Older Person’s Health Status		Adjusted for Caregiver and Household Backgrounds and Older Person’s Health Status and Caregiving Hours		Adjusted for Caregiver and Household Backgrounds and Older Person’s Health Status and Caregiving Hours and Social Support	
Household Handrail Provision	OR	95% CI	OR	95% CI	OR	95% CI	OR	95% CI	OR	95% CI	OR	95% CI
**Panel 1 (1 vs. 2 and 3)** (Ref: no handrail)												
having a handrail in one place	0.56 *	(0.33, 0.94)	0.58	(0.32, 1.04)	0.42 **	(0.22, 0.80)	0.36 **	(0.17, 0.75)	0.30 **	(0.14, 0.66)	0.30 **	(0.14, 0.67)
having handrails in more than one place	3.11	(0.90, 10.82)	3.21	(0.88, 11.69)	1.81	(0.46, 7.13)	2.51	(0.57, 11.00)	5.82 *	(1.05, 32.29)	5.29	(0.98, 28.63)
**Panel 2 (1 and 2 vs. 3)** (Ref: no handrail)												
having a handrail in one place	0.56 *	(0.33, 0.94)	0.58	(0.32, 1.04)	0.42 **	(0.22, 0.80)	0.36 **	(0.17, 0.75)	0.30 **	(0.14, 0.66)	0.30 **	(0.14, 0.67)
having handrails in more than one place	0.82	(0.43, 1.57)	0.72	(0.35, 1.50)	0.45	(0.20, 1.01)	0.47	(0.19, 1.15)	0.43	(0.16, 1.15)	0.45	(0.17, 1.19)
Log-likelihood	−254.01083		−225.27539		−209.40633		−187.1365		−163.71701		−162.48592	
Wald Chi^2^	12.99		70.46		102.2		146.74		193.58		196.04	
Prob> Chi^2^	0.0047		0.0000		0.0000		0.0000		0.0000		0.0000	
Pseudo R^2^	0.0249		0.1352		0.1962		0.2816		0.3715		0.3763	

Note: Caregiver backgrounds- age, sex, marital status, education, working status, relationship to older person(s), duration of being a caregiver, and sandwich carer status; household backgrounds- household wealth index, living arrangement; health status of the older person- cognitive impairment, functional dependency, and behavioral problems. ** *p* < 0.01; and * *p* < 0.05.

## Data Availability

Data are available from the corresponding author upon reasonable request.

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
