# Peer review of "Reducing the Physical Burden of Older Persons’ Household Caregivers: The Effect of Household Handrail Provision"

_ijerph, 2022, doi:10.3390/ijerph19042272_

Round 1
Reviewer 1 Report
Reviewer: summary
The authors conducted a cross-sectional study of caregiver burden in Thailand. They were working with a selected household with older people who need care. Two hundred fifty-four caregivers in households with a dependent older person participated in the study. The study aimed to explore physical burden levels and examine their association with handrail provision among caregivers in older people’s households. The authors reported that the caregivers in older people’s households with one handrail were less likely to experience a high physical burden than those without a handrail, however, there were no statistically significant differences between caregivers living in household with more than one handrail and those of households with no handrails.
Major concerns
The conclusions provided in the study require reflection on the usefulness of its results. The association found between the burden of caregivers living in homes with handrails is remarkable. These results support the authors' main hypothesis and value the appropriate use of handrails in homes where there are dependent people. However, we find contradictory results in no differences in the burden reported by caregivers living in households with more than one handrail compared to households with no handrails. This finding breaks with the hypothesis of the study and the logic of the actual situation that we can find in caregivers' burden. Considering the greater use of handrails, the dependence of older people on their caregivers decreases, the results therefore lack functional logic and do not provide information to the object of study.
Perhaps these results may be distorted by the lack of quantification of caregivers' burden. Quantifying caregivers' burden requires extensive physical and psychological questionnaires and quality of life-related to health. In addition to this, the burden that other daily activities may represent on caregivers, such as the burden at work, the burden of household chores, the burden of commuting from home to the workplace, etc., must be objectively controlled. Furthermore, it is not well maintained whether other household members also act as caregivers since most of the older people live with their family.
Other concerns
The introduction must contain precise information on the state of the art and the hypotheses raised.
Figure 1 is unnecessary.
Figure 2 requires better image quality
The physical caregiver burdens could be present in a table
The Pseudo R2 is relatively low, so the regression explains the results in a low percentage.
The discussion needs a deep rewriting, they must first present the main findings of the study, together with results of comparable studies, as well as an interpretation. This structure must be followed with secondary findings.
Author Response
Point 1: The conclusions provided in the study require reflection on the usefulness of its results. The association found between the burden of caregivers living in homes with handrails is remarkable. These results support the authors' main hypothesis and value the appropriate use of handrails in homes where there are dependent people. However, we find contradictory results in no differences in the burden reported by caregivers living in households with more than one handrail compared to households with no handrails. This finding breaks with the hypothesis of the study and the logic of the actual situation that we can find in caregivers' burden. Considering the greater use of handrails, the dependence of older people on their caregivers decreases, the results therefore lack functional logic and do not provide information to the object of study.
Perhaps these results may be distorted by the lack of quantification of caregivers' burden. Quantifying caregivers' burden requires extensive physical and psychological questionnaires and quality of life-related to health. In addition to this, the burden that other daily activities may represent on caregivers, such as the burden at work, the burden of household chores, the burden of commuting from home to the workplace, etc., must be objectively controlled. Furthermore, it is not well maintained whether other household members also act as caregivers since most of the older people live with their family.
Response 1: Thank you for your comments and insightful suggestions. We have now added the statements for possible explanations of the result in “no differences in the burden reported by caregivers living in households with more than one handrail compared to households with no handrails” and other potential factors that may affect the results and need to be monitored in the Discussion (lines 455–481). Our revised manuscript also includes sentences expressing the value of the findings of this study in the Conclusion (lines 521-528).
Point 2: The introduction must contain precise information on the state of the art and the hypotheses raised.
Response 2: We have now added statements regarding the concepts of home modification on caregiving and caregiver burden (lines 55–70). Also, we added the hypotheses of this study in the Introduction (lines 85–89).
Point 3: Figure 1 is unnecessary.
Response 3: Figure 1 could help clearly define the study site in a single visual to all readers. Since this study collected data in a particular area of Thailand. For that reason, we desire to keep it in the revised manuscript.
Point 4: Figure 2 requires better image quality.
Response 4: The resolution of Figure 2 had been increased (page 4).
Point 5: The physical caregiver burdens could be present in a table.
Response 5: We have added Table 2 to present levels of physical caregiver burden (page 10).
Point 6: The Pseudo R2 is relatively low, so the regression explains the results in a low percentage.
Response 6: We thank the reviewer for pointing this out. We added the sentences to discuss the pseudo R2 value in the Discussion (lines 482–490).
Point 7: The discussion needs a deep rewriting, they must first present the main findings of the study, together with results of comparable studies, as well as an interpretation. This structure must be followed with secondary findings.
Response 7: Thank you for your suggestion. We have rewritten and added more content to discuss the main finding along with references in the Discussion.
Reviewer 2 Report
Overall it is a well written and considered paper. However, I think that the discussion should acknowledge that despite the fine grain nature of the study (visits to actual homes) and the large amount of variables taken into account - the outcomes of the study are still ambiguous and frankly a little underwhelming. That handrails in a household with a person should assist in ADLs and caregiving seems unsurprising (but it is good to have actual numbers based on empirical data). However, the physical burden score of 5 seems to have a high margin of error at 3.85 (i.e. from 1 to 9) so I am not sure how useful it is. That having more than one handrail is as likely to contribute to a high physical burden score as no handrail needs further interpretation - there are many variables but I would start at the income distribution and nationality of caregivers in further investigations. Another addition that would be useful is a description of where the single handrail in a household was (bedroom, bathroom, stairs) and the subsequent effect on caregivers welfare. Lastly, the authors might like to re-word line 267 "demented older people" seems harsh to english-speaking ears - maybe "households with an older person with dementia".
Author Response
Point 1: Overall it is a well written and considered paper. However, I think that the discussion should acknowledge that despite the fine grain nature of the study (visits to actual homes) and the large amount of variables taken into account - the outcomes of the study are still ambiguous and frankly a little underwhelming. That handrails in a household with a person should assist in ADLs and caregiving seems unsurprising (but it is good to have actual numbers based on empirical data). However, the physical burden score of 5 seems to have a high margin of error at 3.85 (i.e. from 1 to 9) so I am not sure how useful it is. That having more than one handrail is as likely to contribute to a high physical burden score as no handrail needs further interpretation - there are many variables but I would start at the income distribution and nationality of caregivers in further investigations. Another addition that would be useful is a description of where the single handrail in a household was (bedroom, bathroom, stairs) and the subsequent effect on caregivers welfare.
Response 1: Thank you for your comments. We added the statements to acknowledge the number of variables that might affect the results in the discussion (lines 486–487). We also added the statements about possible explanations of results in “having more than one handrail is as likely to contribute to a high physical burden score as no handrail” and other potential factors that might affect the results in the Discussion (lines 455–481).
The income distribution and nationality of caregivers have now been considered and added in the limitation of this study and they were recommended for further studies (lines 500–502). However, there are no caregiver welfare and government-subsidized home modification through home care In Thailand.
Point 2: Lastly, the authors might like to re-word line 267 "demented older people" seems harsh to english-speaking ears - maybe "households with an older person with dementia".
Response 2: Thank you the reviewer, we reworded it to ‘households with an older person with dementia’ accordingly (lines 306–307).
Reviewer 3 Report
You have collected data from a population that needs to be studied. All too often the research on caregiver burden comes from high-income countries and high-income populations. The data collection tools you chose are valid and reliable tools.
Now I have included my comments in the attached PDF. The introduction should lead your reader to the need for your study. You start with the assumption that handrails in the home will reduce caregiver burden citing stress model processes as the reasons for handrails reducing caregiver burden. You need to set your research within the body of research on caregiver burden. I think you need to review the research done on home modifications and caregiver burden as well as the research on the caregiver and care-receiver factors related to caregiver burden.
Materials and Methods
What I am looking for in this section is the description of your population, recruitment, ethical approval, tools used for data collection, and then data analysis methods. Thank you for situating your study with maps and descriptions of the region. I have a question about the population. You recruited paid and unpaid caregivers-- "included both informal caregivers (any family member, partner, relatives, friends, or neighbors who had a significant personal relationship with the older household members), and formal caregivers (a provider associated with formal service, whether or not a paid worker or a volunteer); being at least 15 years of age (to gain information based on the caregiving context and caregiver’s perspective); and willing to participate in the study." I would expect some justification for considering them together. Usually, these groups are evaluated separately or sometimes compared. I would expect reporting of each of these in your demographics and at least a comparison in your analysis.
The tools are well described. The CBI is a widely used tool. I would expect reporting of use in other languages and validity and reliability reporting from those other studies. The only reference you cite is the original report. I also would expect how you decided on the cut points for distress, "Therefore, interpretation of the total score was 0 = “no physical 143 burden,” 1–4 = “lower physical burden,” and 5–16 = “higher physical burden.”" Usually authors provide a reference for using particular cut points.
Handrails
Has any other research used this classification for handrails? Why did you include toilet bars or bathtub bars as handrails?
Covariates
You used two stress models to determine the variables that might explain the burden, "Following the Pearlin Stress Process [16], and the Stress Appraisal Model of Yates [17], four domains play important roles in explaining variables of a physical caregiver burden." Is this supported by other research? There is a large body of research on caregiver burden and models of stress and coping. Your analysis methods cannot overcome the selection of variables and interpretation of relevant cut-points for burden and your co-variates.
Analysis
What is the impact on your analysis of this large number of variables? By my calculation, you have 15 variables. Do you need to control for a familywise error rate?
Results
The main problem is the definition and reporting of caregivers. You have recruited formal/ paid and informal/ unpaid and here you are also introducing the term primary. Are you sure that these are comparable groups? Please refer to Pinquart M, Soerensen S. Spouses, Adult Children, and Children-in-Law as Caregivers of Older Adults: A Meta-Analytic Comparison. Psychology and Aging. 2011;26(1):1-14.
Discussion
You claim that burden is high, "According to the level of physical caregiver burden as measured by the CBI, about half of the caregivers of the older-person households in this sample in PhutthamonthonDistrict of Nakhon Pathom Province in 2017 experienced a high burden. If you haven't justified the cut points for your definition of high burden by citing the original authors cut-points or other authors, I am not sure whether this is a reasonable claim.
I also wonder about how you can conclude from your research that "major reason for this is that caregivers in households with handrails in more than one place must be trained to use handrails in different places in order to help older persons carry out specific tasks" It is really important for authors not to overreach in their discussion and conclusions. What can you logically claim based on this quantitative research? As well in your limitations section, you claim, "Finally, with respect to mediating variables, this study focused only on social support as a simple determinant of caregiver burden." If you want to make this claim, I would expect a more explanation of mediating and moderating variables and why particular variables were mediating or moderating variables.
Conclusions
Again I would caution you to decide what can you logically conclude from this study and what is overreaching. As well, you need to decide what belongs in the conclusion. In my view, this entire section does not belong in the conclusions, "This suggests the need for some form of respite care for these caregivers. Some households recruited migrant caregivers from Myanmar, Lao PDR, and Cambodia, and this seemed to be a viable solution (if a relative was not available to provide care). Providing adequate funding for home- or community-based skills training for volunteer, non-Thai, and migrant caregivers should be provided as a stopgap measure to rapidly expand the caregiver labor force. At the household level, establishing age-friendly housing might be one of the significant challenges—and opportunities—to meet long-term care needs for the rapidly growing population of older Thai people." You have introduced many factors related to caregiver burden that were not introduced earlier or discussed in your discussion.
Author Response
Point 1: You have collected data from a population that needs to be studied. All too often the research on caregiver burden comes from high-income countries and high-income populations. The data collection tools you chose are valid and reliable tools.
Response 1: Thank you so much.
Point 2: The introduction should lead your reader to the need for your study. You start with the assumption that handrails in the home will reduce caregiver burden citing stress model processes as the reasons for handrails reducing caregiver burden. You need to set your research within the body of research on caregiver burden. I think you need to review the research done on home modifications and caregiver burden as well as the research on the caregiver and care-receiver factors related to caregiver burden.
Response 2: We are thankful for your suggestion; we have revised by adding statements regarding the concepts of home modification on caregiving and caregiver burden (lines 55–70). Also, caregiver and older people factors related to caregiver burden were added (lines 71–78).
Point 3: Materials and Methods
What I am looking for in this section is the description of your population, recruitment, ethical approval, tools used for data collection, and then data analysis methods. Thank you for situating your study with maps and descriptions of the region. I have a question about the population. You recruited paid and unpaid caregivers-- "included both informal caregivers (any family member, partner, relatives, friends, or neighbors who had a significant personal relationship with the older household members), and formal caregivers (a provider associated with formal service, whether or not a paid worker or a volunteer); being at least 15 years of age (to gain information based on the caregiving context and caregiver’s perspective); and willing to participate in the study." I would expect some justification for considering them together. Usually, these groups are evaluated separately or sometimes compared. I would expect reporting of each of these in your demographics and at least a comparison in your analysis.
Response 3: We do agree with the reviewer. Family members acting as the primary caregivers often experience more substantial pressure, leading to a heavier burden than formal caregivers. However, in Thailand, informal care is the first preference for older people. And in practice, both types of caregivers are often mixed. We have now added the sentence of the reasons why we included both informal and formal caregivers in the part of Materials and Methods (lines 113–117). In addition, we provided the statements to show the proportion of each type of caregiver of this study in the Results (line 310).
Point 4: The tools are well described. The CBI is a widely used tool. I would expect reporting of use in other languages and validity and reliability reporting from those other studies. The only reference you cite is the original report. I also would expect how you decided on the cut points for distress, "Therefore, interpretation of the total score was 0 = “no physical burden,” 1–4 = “lower physical burden,” and 5–16 = “higher physical burden.”" Usually, authors provide a reference for using particular cut points.
Response 4: We have now added the statements reporting the validated CBI in other languages along with their reliability on page 5, lines 162–164. We also provided the statement to show how we determined cut-off scores of this study at lines 170–173.
Point 5: Handrails
Has any other research used this classification for handrails? Why did you include toilet bars or bathtub bars as handrails?
Response 5: Thank you for pointing this out. This study included a bathroom or toilet handrail, but not a bathtub one. We have now added a sentence and references to support our definition of household handrails (lines 184–188).
Point 6: Covariates
You used two stress models to determine the variables that might explain the burden, "Following the Pearlin Stress Process [16], and the Stress Appraisal Model of Yates [17], four domains play important roles in explaining variables of a physical caregiver burden." Is this supported by other research? There is a large body of research on caregiver burden and models of stress and coping. Your analysis methods cannot overcome the selection of variables and interpretation of relevant cut-points for burden and your co-variates.
Response 6: We have added the statement and references to support these two classical models (lines 195–201).
Point 7: Analysis
What is the impact on your analysis of this large number of variables? By my calculation, you have 15 variables. Do you need to control for a familywise error rate?
Response 7: Logistic regression is not multiple comparisons. This is blockwise logistic regression with an increasing number of blocks. The small model is nested in the lasted model so this is not multiple comparisons. However, we appreciated why the reviewer is concerned about examining the p-value from five nested models. For our analysis, if we divided the critical p-value 0.005 by 6 (all six models), we will get 0.008. With this new critical p-value, we have checked and found that our p-value of the last model (OR 0.30** in Model 6) is 0.003 which is still statistically significant.
Point 8: Results
The main problem is the definition and reporting of caregivers. You have recruited formal/ paid and informal/ unpaid and here you are also introducing the term primary. Are you sure that these are comparable groups? Please refer to Pinquart M, Soerensen S. Spouses, Adult Children, and Children-in-Law as Caregivers of Older Adults: A Meta-Analytic Comparison. Psychology and Aging. 2011;26(1):1-14.
Response 8: We thank the reviewer for pointing this out. However, this study did not compare the outcome between the formal and informal groups. We added the reasons why we included both groups in our study (lines 113–117).
Point 9: Discussion
You claim that burden is high, "According to the level of physical caregiver burden as measured by the CBI, about half of the caregivers of the older-person households in this sample in Phutthamonthon District of Nakhon Pathom Province in 2017 experienced a high burden. If you haven't justified the cut points for your definition of high burden by citing the original authors cut-points or other authors, I am not sure whether this is a reasonable claim.
Response 9: We have now clarified the way to determine the cut-off scores of the physical caregiver burden measurement in the part of Materials and Method (lines 170–173).
Point 10: I also wonder about how you can conclude from your research that "major reason for this is that caregivers in households with handrails in more than one place must be trained to use handrails in different places in order to help older persons carry out specific tasks" It is really important for authors not to overreach in their discussion and conclusions. What can you logically claim based on this quantitative research? As well in your limitations section, you claim, "Finally, with respect to mediating variables, this study focused only on social support as a simple determinant of caregiver burden." If you want to make this claim, I would expect a more explanation of mediating and moderating variables and why particular variables were mediating or moderating variables.
Response 10: We thank the reviewer for this comment. We have now deleted those sentences and we added the statements about possible explanations of results in “no differences in the burden reported by caregivers living in households with more than one handrail compared to households with no handrails” and other potential factors that might distort the results in the Discussion (lines 455–481). About the mediating variable, we rewrote and added sentences explaining how a mediating variable works in the caregiver stress/burden process, and why we chose social support of the caregiver as one of the covariates (lines 502–508).
Point 11: Conclusions
Again I would caution you to decide what can you logically conclude from this study and what is overreaching. As well, you need to decide what belongs in the conclusion. In my view, this entire section does not belong in the conclusions, "This suggests the need for some form of respite care for these caregivers. Some households recruited migrant caregivers from Myanmar, Lao PDR, and Cambodia, and this seemed to be a viable solution (if a relative was not available to provide care). Providing adequate funding for home- or community-based skills training for volunteer, non-Thai, and migrant caregivers should be provided as a stopgap measure to rapidly expand the caregiver labor force. At the household level, establishing age-friendly housing might be one of the significant challenges—and opportunities—to meet long-term care needs for the rapidly growing population of older Thai people." You have introduced many factors related to caregiver burden that were not introduced earlier or discussed in your discussion.
Response 11: We deleted the sentence “Some households recruited migrant caregivers from Myanmar, Lao PDR, and Cambodia, and this seemed to be a viable solution…..—to meet long-term care needs for the rapidly growing population of older Thai people”, and rewritten the conclusion by adding more content to reflect the usefulness of the results (lines 521–528). Furthermore, we added the sentence to discuss more the potential factors related to physical caregiver burden in the Discussion (lines 478–481).
Round 2
Reviewer 3 Report
Pg 3 line 113 to line 114 formal carers and formal caregivers. Is one of these supposed to be informal? Other than that you have addressed all my concerns. Thank you very much.
Author Response
Comments:
Pg 3 line 113 to line 114 formal carers and formal caregivers. Is one of these supposed to be informal? Other than that you have addressed all my concerns. Thank you very much.
Response:
Please give me an apology for my carelessness. The correct sentence is 'In Thailand, home-based care is delivered by both informal carers and formal caregivers.'
We have corrected this sentence in the 2nd revised manuscript file (with Trackchanges).
Thank you for pointing out this error.